# Combined but Not Isolated Ingestion of Caffeine and Taurine Improves Wingate Sprint Performance in Female Team-Sport Athletes Habituated to Caffeine

**DOI:** 10.3390/sports9120162

**Published:** 2021-11-27

**Authors:** Raci Karayigit, Alireza Naderi, Bryan Saunders, Scott C. Forbes, Juan Del Coso, Erfan Berjisian, Ulas Can Yildirim, Katsuhiko Suzuki

**Affiliations:** 1Faculty of Sport Sciences, Ankara University, Gölbaşı 06830, Turkey; Ulas.Can.Yildirim@ankara.edu.tr; 2Department of Exercise Physiology, Borujerd Branch, Islamic Azad University, Borujerd 6915136111, Iran; Naderi_a@yahoo.com; 3Applied Physiology and Nutrition Research Group, School of Physical Education and Sport, Rheumatology Division, Faculdade de Medicina FMUSP, University of Sao Paulo, Sao Paulo 01246-903, Brazil; drbryansaunders@outlook.com; 4Department of Physical Education Studies, Brandon University, Brandon, MB R7A6A9, Canada; forbess@brandonu.ca; 5Centre for Sport Studies, Rey Juan Carlos University, 28043 Fuenlabrada, Spain; juan.delcoso@ujrc.es; 6Department of Exercise Physiology, Faculty of Physical Education and Sport Sciences, Tehran University, Tehran 1417935840, Iran; erfan.berjisian@ut.ac.ir; 7Faculty of Sport Sciences, Sinop University, Sinop 57000, Turkey; 8Faculty of Sport Sciences, Waseda University, Totorozawa 359-1192, Japan

**Keywords:** anaerobic capacity, high-intensity exercise performance, ergogenic aids, intermittent exercise

## Abstract

Previous studies have investigated caffeine (CAF) and taurine (TAU) in isolation and combined during exercise in males. However, the potential synergistic effect during high-intensity exercise remains unknown in female athletes. Seventeen female team-sport athletes participated (age: 23.4 ± 2.1 years; height: 1.68 ± 0.05 m; body mass: 59.5 ± 2.2 kg). All participants were habitual caffeine consumers (340.1 ± 28.6 mg/day). A double-blind randomized crossover design was used. Participants completed four experimental trials: (i) CAF and TAU (6 mg/kg body mass of CAF + 1 g of TAU), (ii) CAF alone; (iii) TAU alone; and (iv) placebo (PLA). Supplements were ingested 60 min before a 30-s Wingate Anaerobic Test (WAnT). Heart rate and blood lactate (BL) were measured before and immediately after the WAnT; and ratings of perceived exertion (RPE) were recorded immediately after the WAnT. Peak power (PP) was significantly higher following co-ingestion of CAF+TAU compared to PLA (*p* = 0.03) and TAU (*p* = 0.03). Mean power (MP) was significantly higher following co-ingestion of CAF+TAU compared to PLA (*p* = 0.01). No other differences were found between conditions for PP and MP (*p* > 0.05). There were also no observed differences in fatigue index (FI), BL; heart rate; and RPE between conditions (*p* > 0.05). In conclusion, compared to PLA the combined ingestion of 6 mg/kg of CAF and 1 g of TAU improved both PP and MP in female athletes habituated to caffeine; however; CAF and TAU independently failed to augment WAnT performance.

## 1. Introduction

Intake of energy drinks (EDs) as an ergogenic aid has increased in popularity with a market value of $11 billion in 2018, which is expected to rise to $83.4 billion in 2024 [1]. Most EDs contain caffeine (CAF; 1,3,7-trimethylxanthine) and taurine (TAU; 2-aminoethanesulfonic acid) as the primary purported active ingredients [2,3]. The two primary mechanisms by which CAF improves performance are via antagonism of adenosine receptors in the central nervous system, leading to increases in neurotransmitter release, and potentiation of Na^+^/K^+^ pump activity at the skeletal muscle level, with a potential increase in excitation-contraction coupling [4,5]. TAU, a conditionally essential sulphur-containing amino acid, is commonly ingested in a dose of 1 to 6 g, 60–120 min before exercise and has the potential to enhance exercise performance through a variety of biological processes [6,7]. Within the muscle, TAU enhances sarcoplasmic reticulum Ca^++^ handling in both cardiac and fast twitch skeletal myocytes [8]. Moreover, TAU plays a role as an antioxidant that could improve ATP turnover in the muscle cell, leading to an increase in high intensity exercise performance [7]. In theory, since both CAF and TAU work through different mechanisms (beyond Ca^++^ handling), co-ingestion may further enhance exercise performance than either alone.

Although the combination of ingredients in EDs is purported to increase exercise performance, it is important to note that the dose of CAF in EDs is generally below the dose recommended in the literature (3–6 mg/kg) [9]. CAF (80 mg) and TAU (1 g) co-ingestion, in the doses in a traditional 250-mL serving, failed to enhance sprint cycling performance [10]. Co-ingestion of higher doses of CAF (5 mg/kg) and TAU (4.3 g), improved peak and mean power output during several repetitions of the Wingate anaerobic test (WAnT) in male team-sport players [11]. Interestingly, the ingestion of TAU alone showed a greater performance improvement than co-ingestion of CAF+TAU, suggesting that the interaction in vivo of both supplements may have attenuated the performance effects of TAU [11]. High doses of CAF and TAU co-ingested may work in opposition [10,11], due to the shared mechanisms of Ca^++^ handling and potentiation of ryanodine receptors [7,11]. However, it is unknown whether a high dose of CAF (6 mg/kg) with a low dose of TAU (1 g) (which would limit the potential for opposition) can work synergistically to enhance exercise performance. Furthermore, there have been inconsistent results of CAF and TAU co-ingestion in varying doses provided as an ED on exercise performance. For example, co-ingestion enhanced upper body muscular endurance [12] and time trial performance [13] but had no effect on repeated sprints [14], aerobic endurance [15] and WAnT [16] performance. However, none of these studies isolated the individual effects of CAF and TAU, therefore it is unclear whether co-ingestion enhanced performance compared to CAF or TAU alone. Importantly, none of these studies were performed in women. Benefits reported by studies conducted on men cannot be generalized to women due to physical and physiological differences between sexes such as body size, body composition, hormonal functioning, and oral contraceptive use [17].

Recently, a systematic review and meta-analysis found that CAF intake between 3 and 6 mg/kg improves WAnT performance [9]. The systematic review noted that data in female athletes was lacking. Additionally, several studies failed to assess the efficacy of blinding and often did not report side-effects. Furthermore, habituation to caffeine may be another mediating factor that warrants investigation. One study showed that 3 mg/kg of CAF enhanced peak and mean cycling power during a 15-s WAnT in well-trained female triathletes with habitual daily CAF intake lower than 50 mg/day [18]. However, CAF naïve participants are not truly representative of the athletic population, with 75–90% of athletes using CAF to improve training and competition performance [5]. The effect of CAF on WAnT performance may be diminished with chronic ingestion due to the development of tolerance to caffeine [19]. Therefore, the primary objective was to assess the acute effect of isolated and combined CAF (6 mg/kg) and TAU (1 g) ingestion compared to placebo (PLA) on WAnT performance in female athletes habituated to CAF. We hypothesized that combined ingestion of CAF and TAU would improve WAnT performance in female athletes to a greater extent than the isolated ingestion of CAF or TAU alone, which would all be greater than PLA.

## 2. Materials and Methods

### 2.1. Participants

Seventeen female athletes (age: 23.4 ± 2.1 years, height: 1.68 ± 0.05 m, body mass: 59.5 ± 2.2 kg) participated in this study. All participants were team sport (rugby, football, basketball) players and had previous experience of sprint type activities with a training background of at least four sessions per week for the previous two years. The inclusion criteria were as follows: (a) being involved in regular sprint training; (b) a daily CAF consumption between 3 and 6 mg/kg/day in the previous three months; (c) non-smoker; (d) no injury in the previous six months. All participants were also required not to use creatine, steroids or oral contraceptives. Participants were classified as moderate to high habitual CAF consumers (340.1 ± 28.6 mg/day) according to the classification proposed by Filip et al. [20]. Habitual CAF intake was assessed by an adapted version of the food frequency questionnaire (FFQ) proposed by Bühler et al. [21]. Portion sizes using household measures, were employed to assess the amount of food consumed during a day, a week, and a month. Dietary products rich in CAF including different types of coffee, energy drinks, tea and CAF supplements were listed. A qualified nutritionist estimated the daily CAF intake level for each participant based on the answers to the FFQ. Participants were informed about possible risks of study procedures and provided their written informed consent prior to inclusion into the study. The study was conducted in accordance with the declaration of Helsinki and approved by the Sinop University Human Research Ethic Committee (2021/96).

### 2.2. Experimental Protocol

This double-blind, randomized and crossover study consisted of five separate testing sessions: a familiarization session and four experimental trials with ingestion of isolated CAF or TAU, the combination of these two substances (CAF+TAU) or a placebo (PLA). Peak power (PP), mean power (MP) and fatigue index (FI) were measured during the 30-s WAnT, which was performed 60 min after the oral ingestion of the substances to allow absorption [5,7]. Heart rate (HR; Polar Team 2 telemetric system, Finland) and capillary lactate (BL; Lactate Scout, USA) were measured before and immediately after the WAnT. Ratings of Perceived Exertion (RPE) was measured immediately post-exercise with the 6–20-point scale [22]. All testing took place between 8.00 am and 9.00 am in a fasted state (12 h after the last meal) and trials were separated by at least 48 h to allow substance elimination. Participants were asked to refrain from vigorous exercise and CAF intake from both foods and supplement sources, and to maintain normal dietary patterns in the 24 h prior to testing. The menstrual cycle phase of the participants was not considered because sprint cycling performance has been shown to be improved with CAF to a similar magnitude with CAF supplementation in the follicular, pre-ovulatory and mid-luteal phases [18]. The test procedures are summarized in Figure 1.

### 2.3. Wingate Anaerobic Test

The WAnT was conducted using a specific cycle ergometer (Monark 894E, Peak Bike, Vansbro, Sweden). The seat and handle positions were adjusted for each participant (with a knee angle of approximately 170–175°) in the familiarization session, and they were replicated in the remaining test sessions. Toe clips were used to ensure that the participants’ feet were held firmly in place and in contact with the pedals. Following a 5 min warm-up at 60 W with 5-s sprints without resistance at the second and third minutes, participants were required to pedal from a stationary start with maximum effort for 30 s against a fixed load of 7.5% of participant’s body mass as recommended [23]. Strong verbal encouragement was given and standardized to maintain their highest possible effort throughout the tests. PP and MP were calculated using Monark Anaerobic Test Software (Version 3.3.0.0, Vansbro, Sweden). FI was also calculated via the following formula: ((PP − MP) × 100)/PP [24].

### 2.4. Supplementation Protocol

All supplements were prepared in a powder form, and the doses measured using an analog scale sensitive to 1 mg. The substances were encapsulated in indistinguishable gelatin capsules and ingested with tap water. The capsules contained either 6 mg/kg of CAF (My protein, Manchester, UK), 1 g of TAU (My Protein, Manchester, UK), a combination of 6 mg/kg of CAF and 1 g of TAU, or 300 mg of maltodextrin as a PLA. Supplements were ingested 60 min prior to the start of the WAnT. The CAF dose was chosen based upon recommendations from the International Society of Sports Nutrition position stand on CAF [5]. The dose of TAU was chosen as it is representative of the dose commonly contained in energy drinks currently on the market [1]. The 60 min period was chosen to directly compare with previous studies and commonly account for peak plasma availability of TAU [10] and CAF [5].

### 2.5. Blinding and Side Effect Assessment

Immediately after the experimental trials, participants were asked which condition they thought they had undertaken (i.e., CAF+ PLA, TAU+ PLA, CAF+TAU, or PLA+PLA). They answered two questions: (1) “Can you answer for sure the supplement that you ingested?” (2) “Which supplement do you think you have ingested?” [25]. In addition, participants were asked to outline why they identified the trial as they did [2]. Participants also completed a side-effect questionnaire immediately and 24 h after the end of the experimental trials. The questionnaire included yes/no responses for the presence of eight side-effects commonly associated with caffeine intake [25].

### 2.6. Statistical Analysis

Data are presented as mean ± standard deviations. All data was analysed using the IBM SPSS statistics software for Windows (Version 22.0; IBM Corp., Armonk, NY, USA). PP, MP, FI and RPE were analysed using a one-way analysis of variance (ANOVA) with repeated measures (effect of condition). Two-way repeated measures ANOVA (condition × time) was used to analyse HR and BL. Sphericity was analysed with Mauchly’s test of sphericity followed by the Greenhouse–Geisser adjustment where required. Where any differences were identified, post hoc pairwise comparisons with Bonferroni corrections were conducted. Statistical significance was set at *p* < 0.05. Effect sizes were calculated using 95% confidence interval (CI) and partial eta squared (η^2^), defined as trivial (<0.10), moderate (0.25–0.39) or large (≥0.40).

## 3. Results

### 3.1. Performance in the WAnT

There was a significant effect of condition on PP (F_3,48_ = 6.415; *p* = 0.01; η^2^ = 0.28; Figure 2A). Post hoc analysis revealed that PP with CAF+TAU was significantly higher compared to PLA (+31.9 ± 9.8 W; *p* = 0.03; 95%CI = 2.3–61.4) and TAU (+28.8 ± 9.1 W; *p* = 0.03; 95%CI = 1.3–56.3). There was no difference in PP between CAF+TAU and CAF (*p* = 0.47), CAF and PLA (*p* = 0.19), CAF and TAU (*p* = 0.54) or TAU and PLA (*p* = 0.99). There was a significant effect of condition on MP (F_3,48_ = 3.804; *p* = 0.02; η^2^ = 0.19; Figure 2B). Post hoc analysis revealed that MP in CAF+TAU was significantly higher than PLA (+9.7 ± 2.0 W; *p* = 0.01; 95%CI = 3.4–15.9), but not when compared to TAU (*p* = 0.10) or CAF (*p* = 0.98). There were also no statistically significant differences between CAF and PLA (*p* = 0.99), CAF and TAU (*p* = 0.99) and TAU and PLA (0.99) for MP. FI was not found to be different between conditions (F_3,48_ = 2.343; *p* = 0.08; η^2^ = 0.12; Figure 2C). ICC values were 0.98 in PP and 0.99 in MP.

### 3.2. Blinding and Side Effects

Post experimental trial responses with regard to participants’ awareness of condition indicated that only one participant (5.8%) correctly identified all four conditions, declaring that she felt “more alert” and “more nervous” after ingestion of both CAF + TAU and PLA conditions. Four participants (23.5%) correctly identified CAF+TAU, which is lower than chance (25%). One of these four participants had lower PP in CAF+TAU (608 W) than with CAF (627 W) and TAU (633 W) conditions and equal to PLA (608 W). Five (29%) participants incorrectly identified PLA and thought they were on CAF+TAU. Only two of them had higher PP in PLA (671 and 664 W) compared to CAF+TAU (628 and 661 W), respectively. Increased vigour/activeness was the most declared side-effect in all four conditions measured immediately and 24 h after the experimental trials. These side-effects were higher in the combined CAF and TAU condition (Table 1). 

### 3.3. Blood Lactate, Heart Rate and Ratings of Perceived Exertion

There was no effect of condition (F_3,48_ = 1.046; *p* = 0.36; η^2^ = 0.06) or a condition × time interaction (F_3,48_ = 0.630; *p* = 0.54; η^2^ = 0.03) for BL. However, there was an effect of time (F_1,16_ = 593.492; *p* = 0.01; η^2^ = 0.97), with increased BL post-exercise. There was no effect of condition (F_3,48_ = 2.078; *p* = 0.11; η^2^ = 0.11) or a condition x time interaction (F_3,48_ = 0.314; *p* = 0.81; η^2^ = 0.01) for HR, but there was an effect of time (F_1,16_ = 6901.953; *p* = 0.01; η^2^ = 0.99), with increased HR post-exercise. There was no effect of condition (F_3,48_ = 0.920; *p* = 0.43; η^2^ = 0.05) for RPE (Table 2).

## 4. Discussion

This is the first study to examine the effects of isolated and combined ingestion of CAF and TAU on WAnT performance in team-sport female athletes. Our main findings were that combined, but not isolated, intake of CAF and TAU improved WAnT MP and PP performance compared to PLA. However, co-ingestion was not significantly different than CAF or TAU alone. However, MP in CAF+TAU was superior to TAU alone CAF and TAU alone or co-ingested had no effect on RPE, HR and BL compared to PLA.

The isolated ingestion of TAU did not improve WAnT performance in the present study. Although the ergogenic effects of TAU have been shown, mixed results appear evident with anaerobic exercise [7], and in the current study 1 g of isolated TAU did not improve cycling power during the WAnT in female athletes in a fasted state. In theory, isolated TAU potentiates muscle contractility, which may alter endurance exercise [13] but may be less effective during high intensity exercise [26], as in the WAnT. In support of our findings, power output during repeated sprints (6 × 10 s sprints) did not enhance and even decreased with TAU in female lacrosse players [27]. In contrast, 3.7 g of TAU improved critical power and severe-intensity exercise tolerance in male participants [3]. After exhaustive exercise, type II muscle fibres depleted 25% of its TAU concentration [28] and it is possible that greater benefits of acute TAU ingestion can be seen for experiments including repeated repetitions of the WAnT. Warnock et al. [11] confirmed that approximately 4.3 g of TAU significantly enhanced mean peak power of 3 repetitive WAnT. Collectively, these findings suggest that the isolate ingestion of 1 g of TAU is not effective to increase a single WAnT sprint. However, this dose appears to be effective to enhance aerobic endurance performance [29] and repeated bouts of high intensity exercise. Interestingly, a higher dose of 6 g of TAU does not affect time to exhaustion running [30], however, a high dose of acute TAU may be needed to ensure bioavailability in type II muscle fibers [7,11]. We have previously shown that 6 g but not <4 g of TAU significantly increased peak and mean power during WAnT in female athletes [31]. Thus, the dose here may have been insufficient to improve performance. 

Our results also do not support the isolated ingestion of CAF to improve MP and PP during the WAnT. As previously suggested [5,9], peak and mean power output during WAnT increased with acute CAF ingestion in both male [2] and female [12] athletes. Recent data indicates that the ergogenic benefits of CAF on WAnT performance are of similar magnitude in men and women [32]. The potential ergogenic effect of caffeine has not been corroborated by the present study (Figure 2). Habitual exposure to CAF may induce physiological adaptations leading to a tolerance and blunt the ergogenic effects of acute ingestion [19]. Previous research showing the benefits of CAF on WAnT performance were conducted with low habitual users [2,18,33]. However, not all studies are consistent. Several studies have shown that both endurance [34,35] and strength [36] exercises can be improved with CAF irrespective of low, moderate, or high habitual CAF consumption. It has been suggested that the acute dose of CAF must exceed the daily ingested dose in order to be effective [37]. Nonetheless, the acute dose provided in the current investigation (6 mg/kg) was similar to, but greater than, the mean habitual caffeine intake (5.7 mg/kg/day). Taken together, our findings suggest that the isolated use of caffeine supplementation may not be sufficient to increase WAnT performance in women habituated to caffeine.

The combination of CAF and TAU was the most effective supplementation strategy to augment both MP and PP compared to PLA (Figure 2). Since TAU increases sarcoplasmic reticulum Ca^++^ release [7] and CAF stimulates Na^+^/K^+^-ATPase activity [4], combined ingestion of CAF and TAU may increase the sensitivity of force generating myofilaments in skeletal muscle, in turn, enhancing muscle power output [11,38]. Moreover, in vitro TAU could improve force development of fast twitch muscle fibres in the existence of CAF [39]. Since neither CAF nor TAU in isolation improved these measures, it appears that their synergistic effects were capable of eliciting performance improvements. 

Peak power was not different between isolated CAF and combined CAF-TAU ingestion in this study. However, when an EDs dose of TAU (1 g), which is generally shown to be ineffective in isolation to increase exercise performance [7], was added to 6 mg/kg of CAF, tolerance to the acute CAF intake was overcome. Similarly, 3 mg/kg of CAF and 1 g of TAU significantly improved sprint performance in CAF naïve females [40]. In contrast, Tallis et al. [41] demonstrated physiological concentrations of TAU failed to further potentiate the effects of CAF on short term maximal power output. In addition, 4.3 g of TAU was shown to increase performance during repeated WAnT compared to PLA, but the ergogenic magnitude of this effect diminished when ingested in combination with 5 mg/kg of CAF [11]. Due to the shared molecular target and overlap in CAF and TAU effects on sarcoplasmic calcium release [5,7], speculation can be made that using doses (4.3 g) of TAU that were larger than that of a typical EDs appears to diminish the effect on sprint performance when combined with CAF. This is partially confirmed by the current experiment, as a dose of 1 g of TAU did not interfere but potentiated caffeine ergogenicity in women habituated to caffeine. Therefore, the optimal dose of TAU when combined with 5–6 mg/kg of CAF needs further investigation, and may depend on several factors including participant characteristics, supplement doses and sex differences.

Since the expectation of ingesting TAU or CAF may trigger an ergogenic response known as the PLA effect [25], blinding of participants is critical [9]. Due to low correct treatment identification rates (23.5%) which were distributed similarly in all conditions, it is unlikely that the beneficial effects of combined CAF and TAU ingestion was triggered by expectancy. However, previous CAF consumption experience and habituation level has been suggested as a factor that may alter expectations and possible side effects [42]. Nonetheless, not all studies report such associations, and the current study does not allow us to make any firm conclusions on these hypotheses. Furthermore, the lack of difference in capillary BL, HR and RPE supports previous research [4,12,13,14]. This is congruent with our results which found no difference between conditions, despite the purported moderating effects of TAU and CAF [5,10] on these parameters. 

This study is not without limitations. Plasma CAF and TAU were not assessed and therefore it is unclear whether peak levels occurred 60 min post ingestion. Previous research has shown that peak plasma CAF occurs approximately one-hour post ingestion [5] and TAU peaks at 1–2.5 h [7]. Future research is warranted to determine the influence of timing of these supplements on exercise performance [14]. Furthermore, the menstrual cycles of the participants were not monitored because ergogenic responses to CAF was shown to be of similar magnitude in all three phases [18] but there is no study that examines TAU effects on different menstrual phases. In addition, habitual CAF intake was also assessed by self-report [21]. Lastly, we did not measure plasma neurotransmitter concentration or neuronal activity in skeletal muscle. Therefore, future studies examining mechanisms by which combined CAF and TAU improves anaerobic performance are warranted.

## 5. Conclusions

In summary, the combined ingestion 6 mg/kg of CAF and 1 g of TAU enhanced peak and mean power output during a WAnT in high habitual CAF female athletes. However, the isolated ingestion of these supplements was ineffective. Sport science practitioners and nutritionists could incorporate the use of combined CAF and TAU with athletic cohorts of individuals habituated to caffeine where there may be anaerobic power output demands such as team sports.

## Figures and Tables

**Figure 1 sports-09-00162-f001:**
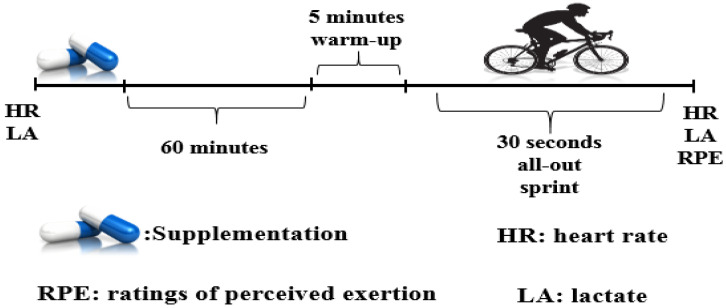
Experimental representation of test protocol.

**Figure 2 sports-09-00162-f002:**
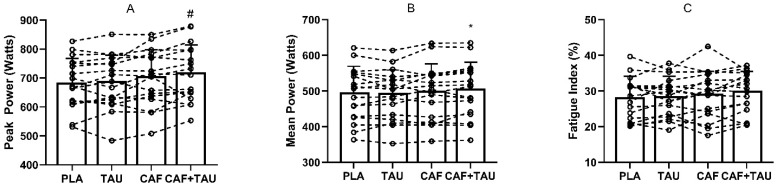
Peak cycling power (**A**), mean cycling power (**B**) and fatigue index (**C**) during the WAnT with the ingestion of taurine (TAU), caffeine (CAF), the combination of CAF+TAU or a placebo (PLA). *: CAF+TAU significantly different from PLA at *p* < 0.05; **#**: CAF+TAU significantly different from PLA and TAU at *p* < 0.05.

**Table 1 sports-09-00162-t001:** Frequency of side-effects with the administration of taurine (TAU), caffeine (CAF), the combination of CAF+TAU or a placebo (PLA) immediately after exercise (+0 HR) and 24 h after (+24 HR).

	PLA		TAU		CAF		CAF+TAU	
Side Effects	+0 HR	+24 HR	+0 HR	+24 HR	+0 HR	+24 HR	+0 HR	+24 HR
Muscle soreness	0 (0%)	0 (0%)	0 (0%)	0 (0%)	0 (0%)	0 (0%)	0 (0%)	0 (0%)
Increased Urine Output	2 (11%)	1 (5%)	0 (0%)	1 (5%)	0 (0%)	2 (11%)	2 (11%)	0 (0%)
Tachycardia and heart palpitations	1 (5%)	0 (0%)	0 (0%)	1 (5%)	3 (17%)	1 (5%)	4 (23%)	0 (0%)
Anxiety or nervousness	3 (17%)	0 (0%)	4 (23%)	1 (5%)	2 (11%)	1 (5%)	3 (17%)	0 (0%)
Headache	0 (0%)	0 (0%)	0 (0%)	1 (5%)	0 (0%)	0 (0%)	1 (5%)	0 (0%)
Gastrointestinal disturbances	1 (5%)	0 (0%)	1 (5%)	1 (5%)	0 (0%)	1 (5%)	2 (11%)	1 (5%)
Increased vigor/activeness	3 (17%)	1 (5%)	2 (11%)	2 (11%)	4 (23%)	2 (11%)	5 (29%)	1 (5%)
Insomnia	1 (5%)	0 (0%)	0 (0%)	0 (0%)	0 (0%)	0 (0%)	1 (5%)	0 (0%)
Total score	11	2	7	7	9	7	18	2

Data are number (frequency) for seventeen female team-sport athletes.

**Table 2 sports-09-00162-t002:** Blood lactate, heart rate and ratings of perceived exertion measured before and after a WAnT with the ingestion of taurine (TAU), caffeine (CAF), the combination of CAF+TAU or placebo (PLA).

	Time Point	PLA	TAU	CAF	CAF+TAU
Lactate (mmol/L)	Before	1.0 ± 0.1	1.0 ± 0.1	1.0 ± 0.1	1.0 ± 0.1
	After *	7.7 ± 1.4	7.5 ± 1.3	8.0 ± 1.7	7.8 ± 1.1
Heart rate (beats/min)	Before	67.2 ± 5.5	66.3 ± 5.0	67.7 ± 5.0	69.0 ± 6.1
	After *	179.8 ± 4.1	179.0 ± 4.6	179.5 ± 6.1	180.2 ± 4.0
RPE (arbitrary units)	After	16.8 ± 1.6	17.0 ± 1.6	16.5 ± 1.5	16.6 ± 1.9

Mean ± SD; *n* = 17 female team-sport athletes. * Effect of time.

## Data Availability

Data is available for research purpose upon reasonable request to the corresponding author.

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
