# Peer review of "Combined but Not Isolated Ingestion of Caffeine and Taurine Improves Wingate Sprint Performance in Female Team-Sport Athletes Habituated to Caffeine"

_sports, 2021, doi:10.3390/sports9120162_

Round 1
Reviewer 1 Report
A well written and robust study in an underrepresented population within this area. I applaud the authors for a great manuscript, as I only have some minor points that I have highlighted below.
Abstract: Clear and concisely written section.
Would just encourage the inclusion of some of the key data to support the results section. Include the actual p values rather than p<0.05.
Introduction:
Well written section again with lots of relevant and important literature. like your section providing more of a rationale with comments on the realistic caffeine intake of athletes to counter research on caffeine naïve individuals.
Ln 52 – can you reword ‘could enhance’. Maybe the use of ‘have the potential to enhance’
Ln 78-81 – I agree that more studies need to be conducted with females. But can you suggest why some of the factors you have identified may influence/effect results?
Method:
Clear method well thought out and good inclusion of blinding and side effect questions. Very minor points below.
Ln 126 – provide reference for the use of RPE scale
Ln 129-130 – you have stated that participants were asked to refrain from caffeine in the 24 hours prior to exercise, but to consume their normal dietary patterns? Would caffeine consumption fall into normal dietary patterns, minor point and I am probably being picky here…sorry.
Results:
Results communicated well in test. Relevant detail from the statistical analysis process.
Think figure 2 could be improved:
Figure 2: can you try and keep these consistent. You have used individual data points on two figs but not the third. Can these be added. How would SD bars look on these? Maybe messy due to the individual points. Can you remove the figure title that you have on each one, not needed and identified with a, b or c and links to the figure label. Also remove the boarders.
Like your section 3.2. good addition.
Discussion:
Great section highlighting all the key points, and supported with relevant literature throughout. Solid limitations and conclusion to finish. Minor points below:
Ln 233-234 – can you reword this and separately mention PP and MP, as there is sig diffs between combined and singular ingestion for MP and you mention in parenthesis.
Paragraph starting at Ln 237 – good detail from the literature for TAU. But early in this paragraph you said that ‘ergogenic effects of TAU have been shown in males, [7],’.
Ln 273 – you mention the use of 9 mg/kg. But it may be prudent to highlight the potential of further side effects with such large doses, and that previous research have not shown improvements with greater amounts i.e. Graham and Spriet (1995) and Bruce et al. (2000). Although your point on getting above their habituated intake is valued. Large range of habituation within the Graham and Spriet paper and not clear within the Bruce paper.
Ln 300 – reword ‘Because’ – maybe use ‘Since …’
Author Response
Thank you very much for you spend your time and give us suggestion and make our paper better. Our corrections are below.

Reviewer 2 Report
Congratulation! Karayigit et al. investigated the synergic effect of caffeine and taurine on the Wingate test in female university athletes using a double-blind test.
To improve this article, the Reviewer suggests some minor comments.
1) Authors: Please delete and *It seems an error.
2) Line66-68: Please kindly add information about high and low
3) Line 82: The Reviewer believes describing the situation how much male and female; this would make a robust the sentence.
4) Line 101: Which kind of team sports are included?
5) How to recruit the participants (for example, poster, direct)
6) How many participants were excluded due to out-of-range caffeine consumption.
7) Line 145: I am not sure official rule, but could you delete Bar-Or O? it looks wired for me just show family name without a title.
8) Figure 2. If you could time the course of the figure, please show it. It makes the reader understand how fast to reach the peak and whether mean power depends on peak power or not.
Author Response
Firstly thanks for your suggestions that get our paper's quality higher. Our revisions are in attach. Regards.
